# Exploring and Evaluating Personalized Models for Code Generation

## Abstract

Large Transformer models achieved the state-of-the-art status for Natural Language Understanding and are increasingly the baseline architecture for source code generation models. Transformers are usually pre-trained on a large unsupervised corpus, learning token representations and transformations relevant to modeling generally available text, and then fine-tuned on a particular task of interest. While fine-tuning is a tried-and-true method for adapting a model to a new domain, for example question-answering on a given topic or a source code generation model, generalization remains an on-going challenge. Here we explore the ability of various levels of model fine-tuning to improve generalization by personalized fine-tuning. In the context of generating unit tests for Java methods, here we evaluate learning to personalize to a specific project using several methods to personalize transformer models for unit test generation for a specific Java project. We consider three fine-tuning approaches: (i) *custom* fine-tuning, which allows all the model parameters to be tuned; (ii) *lightweight* fine-tuning, which freezes most of the model's parameters, allowing a tuning of the token embeddings and softmax layer or the final layer alone; (iii) *prefix* tuning, which keeps language model parameters frozen, but optimizes a small project-specific prefix vector. Each of these techniques offers a different trade-off in total compute cost and prediction performance, which we evaluate by code and task-specific metrics, training time, and total computational operations. We compare these fine-tuning strategies for code generation and discuss the potential generalization and cost benefits of each in deployment scenarios.

## 1 Introduction

It is well-known that even the best models can fail to generalize properly to new domains, and even to new users of said models. For example, a model trained to answer questions in general may not answer StackOverflow questions as well as the questions in the training domain, or a software developer in an Enterprise environment with private code may have libraries and attribute name which differ from public source code used to train a code synthesis model.

The current dominant paradigm in Natural Language Processing (NLP) modeling is to pre-train a large transformer model (Vaswani et al., 2017a) on a large corpus and then fine-tune it on a particular task of interest. For example, a question-answering (Q&A) model is generally first pre-trained on a large corpus of textual data for the specific language (*e.g.,* Wikipedia, and news articles in English), then fine-tuned on a task-specific dataset of paired questions and corresponding answers. The pre-training process aims at learning semantic vector representation of the language and words, while the fine-tuning process specializes the model for a specific domain.

Transformer models are also increasingly the baseline architecture used for code generation tasks, such as writing methods from natural language description (Clement et al., 2020; Austin et al., 2021; Chen et al., 2021), or generating test cases from the focal method under test (Tufano et al., 2021). Similarly for NLP tasks these models are pre-trained on a large corpus of natural text and publicly available source code and then fine-tuned on a specific code-related task. Further, these models also may not generalize to new domains of interest, and can benefit from task or even user-specific fine-tuning, here called customization or personalization. Customization is particularly relevant for code generation models since it provides several benefits:

- allows fine-tuning on source code data that may not be available when training a base model (*e.g.,* private repositories or internal codebases), enabling improved overall performances on codebases with proprietary dependencies and code styles;
- the opportunity to improve data privacy by considering private or sensitive data only during the customization process on the client side;
- the opportunity to reduce deployment cost as customized models can offer better user performance without increasing model size.

Custom models can provide clear benefits to users and model providers. We envision serving tens or hundreds of thousands of custom models, but doing so presents several logistical hurdles, including the costs of training, storing, and loading these models into GPU memory for inference. Worse, memory costs will only be exacerbated when working with ever larger and more powerful models.

For these reasons, we investigate several customization approaches, some of which can dramatically reduce the memory footprint and amortized computational cost introduced by custom models. Specifically, we consider three fine-tuning approaches: (i) *custom* fine-tuning, which allows all the model parameters to be tuned; (ii) *lightweight* fine-tuning, which only optimizes the token embedding representations or the final softmax layer; (iii) *prefix* tuning, which keeps language model parameters frozen, but optimizes a small project-specific vector prefix.

In our extensive empirical evaluation we found that all the customization strategies lead to significant model improvements on a target project in terms of both intrinsic and task-specific metrics. While there is no unambiguous winner among the customization strategies, each approach can provide specific benefits in particular deployment scenarios. This paper provides insights on these customization strategies, their benefits and drawbacks, as well as providing guidelines and suggestions on which one to use based on the training cost, memory and storage, number of users, and deployment scenarios.

## 2 APPROACH

This section describes the proposed customization approach for code generation models. We begin by formally defining the customization process, then we provide details for each of the fine-tuning strategies.

### 2.1 CUSTOMIZATION PROCESS

We use the term *customization* to refer to the process of fine-tuning a model $m$, previously trained on a generic dataset for a task $t$, with the goal of improving its performance on a specific dataset $p$. The performance of a machine learning model $m$ on a dataset $p$ is measured by one or more evaluation functions $f(m, p)$, where $f$ can be either a maximization (*e.g.,* BLEU, top-k accuracy) or minimization (*e.g.,* perplexity) function. The customization process is designed to modify the trainable parameters of the model $m$, obtaining the model $m'$, such that the performance of $m'$ on $p$ is better than what was attained by $m$. Specifically, $f(m', p) > f(m, p)$ for maximization functions, or $f(m', p) < f(m, p)$ for minimization functions.

In this work, $m$ is an encoder-decoder transformer model, $t$ is a code generation task, and $p$ is a target software project to which we intend to customize $m$.

### 2.2 CUSTOM FINE-TUNING

Custom fine-tuning is the most straightforward customization approach. The model to be customized is taken as is and trained on a selected project. All parameters are trainable during this process. Figure 1a shows the model during fine-tuning, where all the parameters from the encoder and decoder blocks, as well as embeddings and output layers can be modified.

### 2.3 LIGHTWEIGHT FINE-TUNING - EMBEDDINGS AND OUTPUT LAYER (L-EO)

Fully fine-tuning a model for every project or user may be prohibitive in terms of storage and memory costs. As a result, we explore ways to mitigate these costs by reducing the number of parameters that vary from one custom model to another. In our lightweight fine-tuning experiments, we achieve this

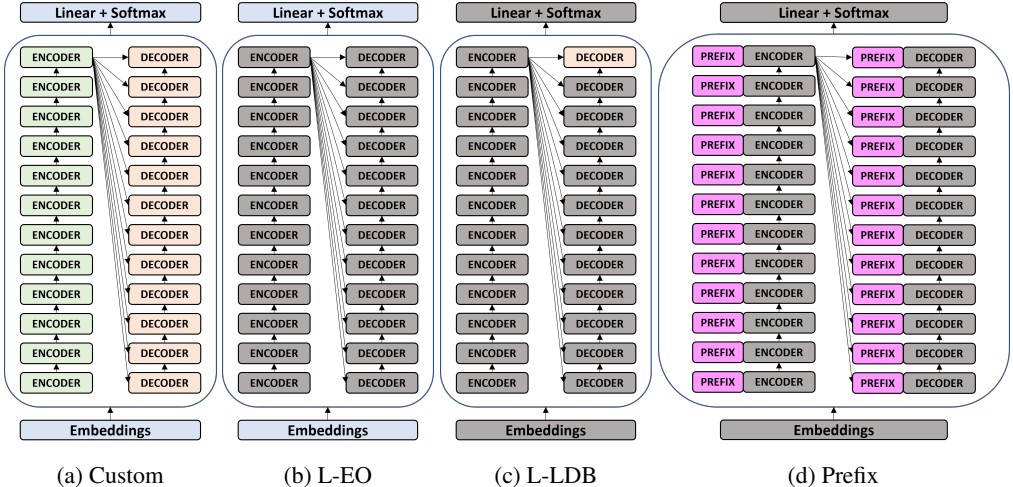

Figure 1: Overview of the Customization Approaches - Transformer models during fine-tuning, where the frozen parts of the model (not trainable) are displayed in gray: (a) Custom fine-tuning modifies all parameters during training; (b) L-EO trains only the embedding and output layers; (c) L-LDB allows to train only the parameters of the last decoder block; (d) Prefix tuning adds a trainable prefix to the encoder and decoder blocks.

by freezing most parameters in the baseline model, and only keeping a small subset trainable. Figure 1b shows the Lightweight fine-tuning - Embeddings and Output Layer (L-EO) design, where most of the model parameters are frozen (displayed in gray), and we allow only the embedding and output layers parameters to be fine-tuned, following the approach in Lu et al. (2021).

## 2.4 LIGHTWEIGHT FINE-TUNING - LAST DECODER BLOCK (L-LDB)

In this lightweight fine-tuning strategy, shown in Figure 1c (L-LDB), most of the model's parameters are kept frozen, while only the parameters in the last decoder block are trainable, this includes: self-attention, encoder-decoder attention, layernorm and feedforward layers. This design decision of training only the last decoder block is motivated by experimental results analyzing the model's parameter changes during custom fine-tuning. Figure 2 reports the average absolute changes, during fine-tuning, in the parameters belonging to different Encoder and Decoder blocks for a BART model. We observe that, as we go through the transformer model, the average change in parameter values tends to increase, with the last decoder block showing the highest changes in parameter values. As a result, we hypothesize that it could be sufficient to tune the last decoder block and obtain performance improvements similar to the fully custom fine-tuned model.

## 2.5 PREFIX TUNING

Prefix tuning was first introduced by Li and Liang (2021), with the goal of fine-tuning a general model for different tasks. The technique concatenates a sequence (prefix) of virtual tokens (trainable parameters) to the front of the input of every encoder and decoder block. In our context, the intuition behind this approach is that the prefix embeds the properties of a specific project, which allows the model to generate customized responses for that repository. Practically, we set the prefix length to 200 tokens, and thus with an embedding size of 1024, this gives a total of $1024 \times 200 \times 24 \times 2 \approx 10M$ trainable parameters. The prefix is initialized to the most frequent words in the repository for which the model is customized.

## 2.6 TRAINABLE PARAMETERS DURING FINE-TUNING

Table 1 provides an overview of the number of total and trainable parameters involved in each customization process, in the case of a BART Transformer model with 406M parameters. Custom fine-tuning allows to train 100% of the 406M available parameters in the model. During L-EO finetuining, instead, only 13% (53M) parameters are trained. The L-LDB fientuning reduces the

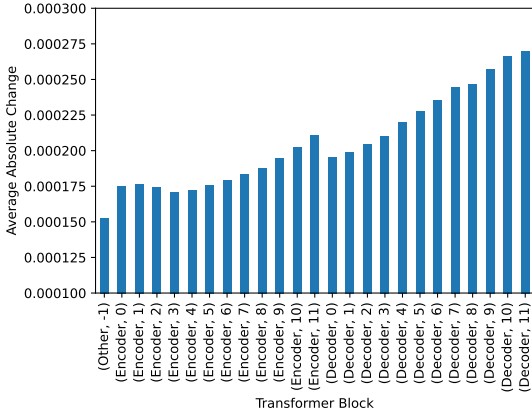

Figure 2: This figure shows the total average parameter change after fine-tuning to a new project domain, showing that the largest parameter changes occur in deeper parts of the model. This motivates our choice to try only fine-tuning the later layers of the model.

number of trainable parameters to 4.2% (17M). Finally, Prefix tuning has the lowest number of trainable parameters, only 2.4% (10M) of the total, but these are additional parameters added to the model, which reaches a total of 416M.

| Customization Process | Parameters | |
|---|---|---|
| | Total | Trained |
| Custom | 406M | 406M (100%) |
| L-EO | 406M | 53M (13%) |
| L-LDB | 406M | 17M (4.2%) |
| Prefix | 416M | 10M (2.4%) |

Table 1: Comparing the number of trainable parameters in each fine-tuning method.

## 3 EXPERIMENTAL DESIGN

The goal of our experimental design is to investigate whether custom models outperform the baseline model, leading to performance improvements in terms of intrinsic metrics ($RQ_1$), as well as extrinsic task-specific metrics ($RQ_2$). Next, we analyze and compare the different customization approaches in terms of training and compute costs ($RQ_3$) as well as model size and required storage for deployment.

In our case study, we chose Unit Test Case generation as our code generation task $t$, and *AthenaTest* by Tufano et al. (2021) as our baseline model $m$, which is a BART transformer model pre-trained on source code and English, and fine-tuned on Java unit test generation. The task is modeled as a translation task, where the input is a focal method (*i.e.,* method under test), and the output is a test case which tests the focal method's correctness. We randomly sample 20 projects from the test set, each of those representing the dataset $p$ on which a custom model is fine-tuned. Specifically, for each project $p$, we start from the baseline model $m$ and fine-tune four different custom models according to the four proposed fine-tuning strategies. For each project and fine-tuning strategy (*e.g.,* L-EO), we fine-tune and evaluate the models using 4-fold cross-validation. The models are trained until the best validation loss is reached, independently for every fold, every repository, and every customization approach. In total, we fine-tune and evaluate $20(projects) \times 4(approaches) \times 4(folds) = 320$ models.

### 3.1 DATASET

Table 2 reports information about the 20 GitHub repositories sampled from the test set, which will be used to customize our models. The table shows (i) the Project ID, which will be used in the paper to

reference a specific project; (ii) the project name; (iii) the project size in terms of disk usage; (iv) the popularity of the project in terms of number of stars obtained on GitHub; (v) and the dataset size, which corresponds to the number of data points for the unit test generation task (*i.e.,* pair of focal method and test case). The list of projects represent a diverse set of repositories with different size, domain, and popularity. They span from small personal projects (*e.g.,* `Tutorials` with 6 stars), to open source projects developed by large organizations such as Apache and Google.

| Project ID | Name | Project Size (MB) | Stars | Dataset Size |
|---|---|---|---|---|
| 26644682 | Talend Data Prep | 68.8 | 56 | 651 |
| 40735368 | GeoTools | 62.4 | 8 | 653 |
| 107330274 | Titus Control Plane | 36.0 | 302 | 660 |
| 52972024 | Smart Actors | 57.8 | 22 | 704 |
| 9714608 | Arakhnê Foundation Classes | 17.9 | 13 | 753 |
| 60701247 | Android Plugin for IntelliJ IDEA | 1026.7 | 716 | 754 |
| 14550159 | EverRest | 5.3 | 24 | 761 |
| 9278888 | Brave | 18.8 | 2084 | 787 |
| 66940520 | DHIS 2 | 118.1 | 211 | 862 |
| 33645537 | Tutorials | 34.4 | 6 | 878 |
| 62253355 | Mobi | 62.6 | 35 | 986 |
| 155883728 | OakPAL | 15.0 | 9 | 1005 |
| 4710920 | Apache Dubbo | 36.1 | 36231 | 1058 |
| 29603649 | Wilma | 6.7 | 40 | 1074 |
| 42949039 | Herd | 227.2 | 127 | 1249 |
| 1381673 | Drools | 176.7 | 3908 | 1394 |
| 1244027 | ModeShape | 131.1 | 212 | 1550 |
| 73948366 | AthenZ | 38.8 | 639 | 1920 |
| 660443 | Chemistry Development Kit (CDK) | 214.8 | 305 | 2591 |
| 87849739 | Eclipse Ditto™ Project | 52.5 | 311 | 2842 |

Table 2: Dataset - Projects used for customization

## 3.2 RQ$_1$: Intrinsic Evaluation Metrics

**RQ$_1$: Do custom models obtain better performances on intrinsic metrics, such as BLEU and perplexity, w.r.t. the baseline?** To begin, we investigate how the different model customization approaches described in Sec. 2 score on intrinsic metrics such as BLEU and perplexity. All approaches entail fine-tuning the baseline model to the dataset of a specific project, with the choice of parameters being tuned depending on the approach taken. The four variants are trained independently until the best validation loss is achieved. We report the BLEU4 score and the mean perplexity per token on the test fold, for all the 20 projects. Next, we perform statistical tests to investigate whether the observed differences between the baseline and custom models are significant, as well as differences among the customization approaches. Specifically, we rely on the Kruskal-Wallis test, a non-parametric statistical test.

## 3.3 RQ$_2$: Task-specific performances

**RQ$_2$: Do custom models improve on performance metrics specific to unit test generation?** We want to investigate how the different customization approaches compare with respect to the downstream task of generating unit tests. Beyond BLEU score and perplexity, we would like to see if custom models can produce the correct target code, how closely their unit tests mimic the repository style, or even if they can perfectly match the desired output.

- *Perfect Matches*: We compare the model's output string with the target developer-written unit test. If the two strings are identical, this is considered a perfect match. We do not take spacing and indentation into account as we are using a Java dataset (where indentation is not required). We report the proportion of perfect matches among the top 5 model predictions.

- *Abstracted Code Matches*: We pass the model output and target output through the src2abs tool (Tufano, 2018), to obtain an abstracted version, masking variable names, method names, etc. We also do not distinguish between different objects of the same type.

- *Coding Style*: For each project's custom model, we would like to determine how closely the model learns the developer's personal programming style and preferences. To this end, we extract the collection of all identifiers (*i.e.,* variables and functions' names) from the unit tests written by the developer as well as those generated by the models. We then pass these text outputs through a tf-idf vectorizer and compute the cosine similarity between them. This allows us to compare the developer's and the models' word usage. We examine the similarity between the developer's unit tests and the baseline and custom models generated tests. This scores the vocabulary similarity of the unit tests with the model generated code.

## 3.4 RQ$_3$: Training cost comparison

**RQ$_3$: Given the same amount of compute, which custom models achieve the biggest performance improvement?** Since our four training regimes tune a different number of parameters, simply comparing the training time or number of optimization steps to reach the best validation loss may not be appropriate. For a model with $N$ parameters, we approximate the computation cost of a forward pass to be $C \approx 2N$ floating point operations per training token, with an additional correction for embedding layers. The backward pass takes roughly twice the amount of compute, but it is unnecessary for layers that are frozen. For additional details, we refer to Table 1 in Kaplan et al. (2020). We report the resulting compute in petaFLOPS-seconds.

## 4 RESULTS

### 4.1 RQ$_1$: INTRINSIC EVALUATION METRICS

Table 3 presents the results of custom models in terms of the intrinsic metrics: BLEU and perplexity. Specifically, for each project, we report the average BLEU and perplexity over the four folds, achieved on the test set by the different customization strategy, as well as the baseline model. We observe notable improvements in both metrics for every project w.r.t. the baseline, with BLEU going from 16.1 achieved by the baseline model to 36-38 by custom models.

The statistical tests reported in Table 4 confirm that the improvement observed by the four customization techniques are statistically significant w.r.t. the baseline ($p <$ 1e-7). However, we do not observe statistical significance in the differences among the customization strategies, meaning that, in terms of intrinsic metrics performances, the differences are within margin of error.

### 4.2 RQ$_2$: TASK-SPECIFIC PERFORMANCES

The results in terms of task-specific performances are presented in Figure 3. The plot 3a shows the top-k accuracy for perfect matches (solid line) and abstracted matches (dotted line), aggregated over the 20 projects. The baseline model outputs the same code structure (abstracted) in roughly 3% of all cases, and virtually never produces the exact target output (<1%). Moreover, its performance does not improve as we consider more predictions. All customization processes show significant improvement compared to the baseline. Specifically, these improvements are observed for every single project (full results will be available on our online appendix). Customized models produce the correct code structure as their top prediction in ∼13-14% of instances, and a perfect match in ∼4-6% of cases. They also tend to improve as we consider their top 5 predictions. Between the different customization processes, Custom consistently performs the best, closely followed by Prefix and L-LDB. When considering abstracted code matches, these three approaches are nearly identical. L-EO, however, performs slightly worse than the others.

Plot 3b shows the distribution of tf-idf cosine similarity computed between identifiers used in the developers' written tests and the models' generated outputs. We observe that the distribution for custom models is skewed towards the higher values of cosine similarity. This result demonstrates that custom models tend to use variable and function names that are more similar to what developers used in their own tests.

| Project | BLEU4 | | | | | Perplexity | | | | |
|---------|-------|------|------|-------|--------|-------|-------|-------|-------|--------|
| | Base | Cust. | L-EO | L-LDB | Prefix | Base | Cust. | L-EO | L-LDB | Prefix |
| 26644682 | 14.1 | 32.9 | 31.6 | 31.9 | **34.0** | 1.275 | 1.212 | 1.208 | 1.238 | **1.197** |
| 40735368 | 18.5 | **30.7** | 29.0 | 29.4 | 29.1 | 1.276 | **1.186** | 1.197 | **1.186** | 1.194 |
| 107330274 | 14.8 | **38.0** | 35.0 | 35.9 | 35.7 | 1.273 | **1.160** | 1.168 | 1.164 | 1.175 |
| 52972024 | 10.2 | 31.8 | **33.2** | 32.2 | 30.1 | 1.271 | 1.142 | 1.146 | **1.135** | **1.135** |
| 9714608 | 14.7 | **41.0** | 38.1 | 40.4 | 40.2 | 1.263 | 1.155 | 1.145 | 1.150 | **1.138** |
| 60701247 | 10.8 | **28.9** | 24.4 | 25.9 | 26.6 | 1.267 | 1.187 | 1.190 | **1.172** | 1.176 |
| 14550159 | 20.0 | **49.5** | 47.2 | 46.6 | 46.4 | 1.245 | 1.121 | 1.122 | **1.116** | 1.124 |
| 9278888 | 17.3 | 46.8 | 44.5 | 47.2 | **47.8** | 1.272 | **1.137** | 1.152 | 1.138 | 1.140 |
| 66940520 | 17.4 | **37.9** | 33.9 | 35.5 | 37.7 | 1.264 | 1.154 | 1.163 | 1.154 | **1.150** |
| 33645537 | 17.0 | 30.4 | 31.2 | **32.0** | 31.0 | 1.264 | 1.231 | 1.200 | **1.192** | 1.211 |
| 62253355 | 14.7 | **48.0** | 45.7 | 47.3 | **48.0** | 1.292 | **1.113** | 1.114 | 1.114 | 1.116 |
| 155883728 | 13.7 | **41.3** | 37.5 | 39.3 | 39.5 | 1.238 | **1.132** | 1.148 | 1.146 | 1.140 |
| 4710920 | 28.2 | **39.1** | 38.1 | 38.8 | 38.6 | 1.218 | 1.161 | 1.162 | 1.167 | **1.160** |
| 29603649 | 19.1 | **58.4** | 54.9 | 56.6 | 56.8 | 1.266 | **1.096** | 1.110 | 1.099 | 1.098 |
| 42949039 | 17.0 | **38.2** | 37.7 | 37.5 | 37.3 | 1.238 | 1.154 | 1.152 | 1.154 | **1.148** |
| 1381673 | 14.3 | **33.3** | 29.3 | 30.9 | 30.8 | 1.261 | **1.133** | 1.152 | 1.138 | 1.138 |
| 1244027 | 19.6 | **30.1** | 29.7 | 30.0 | 30.0 | 1.244 | **1.142** | 1.160 | **1.142** | 1.150 |
| 73948366 | 12.0 | 33.1 | 31.8 | **34.0** | 33.6 | 1.267 | 1.161 | **1.157** | 1.159 | 1.164 |
| 660443 | 15.0 | 34.0 | **37.2** | 36.5 | 34.3 | 1.281 | 1.180 | 1.170 | **1.169** | 1.177 |
| 87849739 | 13.4 | 45.1 | 47.0 | **48.9** | 46.8 | 1.259 | 1.138 | 1.136 | **1.124** | 1.144 |
| Average | 16.1 | **38.4** | 36.9 | 37.8 | 37.7 | 1.262 | **1.153** | 1.158 | **1.153** | 1.154 |

Table 3: The BLEU score and perplexity for the customization methods evaluated on the 20 projects in our test set.

| | BLEU4 | | | | | Perplexity | | | | |
|--------|-------|-------|-------|-------|--------|-------|-------|-------|-------|--------|
| | Base | Cust. | L-EO | L-LDB | Prefix | Base | Cust. | L-EO | L-LDB | Prefix |
| Base | - | **3e-08** | **3e-08** | **3e-08** | **3e-08** | - | **3e-08** | **3e-08** | **3e-08** | **3e-08** |
| Cust. | **3e-08** | - | 0.4 | 0.7 | 0.7 | **3e-08** | - | 0.5 | 0.9 | 0.9 |
| EO | **3e-08** | 0.4 | - | 0.5 | 0.7 | **3e-08** | 0.5 | - | 0.5 | 0.5 |
| LDB | **3e-08** | 0.7 | 0.5 | - | 0.9 | **3e-08** | 0.9 | 0.5 | - | 0.8 |
| Prefix | **3e-08** | 0.7 | 0.7 | 0.9 | - | **3e-08** | 0.9 | 0.5 | 0.8 | - |

Table 4: Kruskal-Wallis Test p-values testing the significance of the pairwise hypothesis that one customization method is superior than another. Custom strategies are significantly better than baseline.

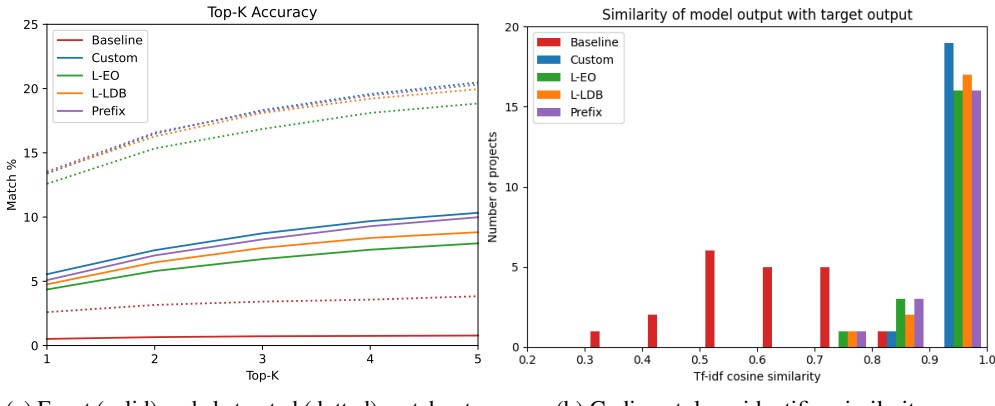

(a) Exact (solid) and abstracted (dotted) match rate     (b) Coding style as identifier similarity

Figure 3: Task-specific metrics (a) custom models outperform the baseline in terms of perfect matches (solid line) and abstract matches (dotted line); (b) custom models generate code that uses identifiers (*i.e.*, variable and function names) that are more similar to the project codebase.

### 4.3 RQ₃: TRAINING COST COMPARISON

For each customization process, we plot validation loss as a function of compute, as defined in section 3.4. The results are presented in Figure 4, where the light lines represent the validation loss curve for each individual project and fold, while the bold line represents the average for each custom strategy. First note that Custom achieves very large gains during the first epoch, as evidenced by the fact that its validation loss starts much lower than L-EO and L-LDB. Custom also outperforms other customization processes when given a limited amount of compute. However, we observe that beyond a certain amount of compute, Custom and L-LDB tend to achieve similar performances. In contrast, L-EO starts at the same validation loss as L-LDB but converges much slower to the best loss, requiring 2-3 times as much compute.

Since the prefix parameters suffer from poor initialization, Prefix is the most expensive customization process. To overcome this problem, it is possible to first train the prefix on a large generic dataset. Then, given proper hyperparameter tuning, it is possible to substantially cut down compute cost for customizing the prefix.

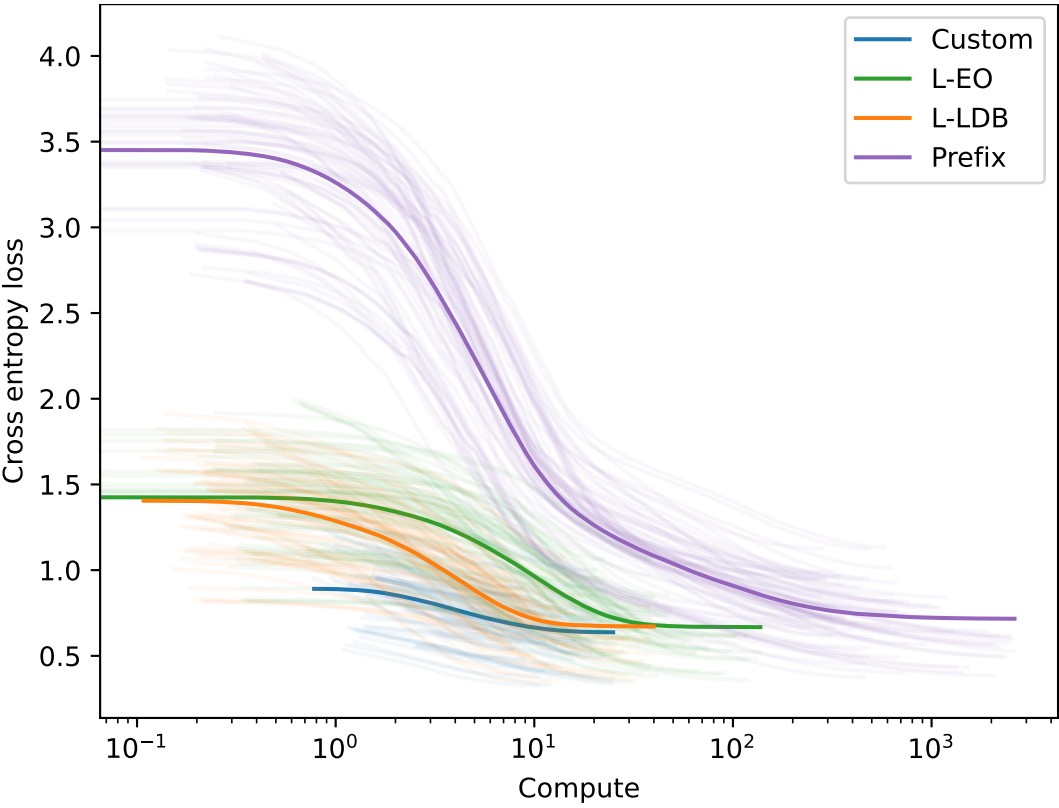

Figure 4: Validation Loss vs Compute (PF-seconds) - Light lines represent the validation loss curve for each individual project and fold, while the bold line represents the average for each custom strategy. Custom is the most efficient, lightweight approaches require slightly more compute to reach a comparable validation loss, while prefix is the least efficient, suffering from poor initialization.

## 5 DISCUSSION & LESSONS LEARNED

The four customization strategies considered in this work are effective in improving a code generation model's performances on a given software project. Specifically, all custom models significantly outperform the baseline in terms of intrinsic metrics (*i.e.,* BLEU and perplexity) as well as task-specific metrics (*i.e.,* abstract and raw matches). While the differences among the customization approaches are not significant (no clear winner), each strategy offers specific advantages in different circumstances and deployment scenarios.

***Custom*** fine-tuning achieves the overall best performances and the customization process is relatively fast and efficient. This is somewhat expected, since this customization strategy allows all the model's parameters to be tuned on the specific project. This characteristic also leads to the major disadvantage of this approach: each custom model is an entire copy of the original model. Storage and inference costs could become prohibitive when serving many users with personalized custom models.

***Lightweight*** fine-tuning achieves good results while training fewer parameters. This allows to serve potentially many users with custom models which can be stored and loaded efficiently. Specifically, L-LDB trains fewer parameters than L-EO, however the latter could allow to deploy the embedding and output layers on the user side, with a privacy-preserving focus.

***Prefix*** fine-tuning trains the lowest number of parameters (only 2.4% for a BART model), while improving over the baseline. However, it increases the total number of parameters of the model (prefixes are additional virtual tokens) and requires more compute time to achieve good performances, mostly due to the prefix initialization problem. On the bright side, this strategy allows to batch together requests from different users (with different prefixes), which can be processed by a single model, generating personalized outputs.

## 6  RELATED WORK

This work is related to two areas of the existing literature: neural source code generation and model personalization. Neural code generation has generated an intense recent interest in NLP, using Transformer models Vaswani et al. (2017b) in particular for code completion Svyatkovskiy et al. (2020; 2019); Clement et al. (2020); Raychev et al. (2014); Bruch et al. (2009); Brockschmidt et al. (2018), code synthesis from examples Chen et al. (2018), natural language to code Clement et al. (2020); Chen et al. (2018); Austin et al. (2021), code feature summarization Liu et al. (2021); Moreno et al. (2013); Scalabrino et al. (2017); Wan et al. (2018); Alon et al. (2018); Moreno et al. (2014), code search Husain et al. (2019); Feng et al. (2020), unit test generation Tufano et al. (2021) and even bug fixing Drain et al. (2021) and detection Zhai et al. (2020). This paper naturally is an extension and evaluation of personalized unit test generations as studied by Tufano et al. (2021), and an important contribution to the understanding optimization in a deployment scenario.

Much of the previous literature on personalized models focuses on client-side training to keep data on device Shor et al. (2019); Popov et al. (2018), and most work is in the domain of search query completion Jaech and Ostendorf (2018), natural language completion Popov et al. (2018), or even automated speech recognition Shor et al. (2019). Naturally this work extends the domain of evaluation beyond natural language tasks and into the software engineering domain. This paper does not evaluate methods for client side training with restricted resources, however, as the most powerful large language models which enable state of the art code synthesis have 10-100 million parameters. At the time of writing such large models cannot be executed in a reasonable amount of time on most consumer laptops. We leave to future work extending these studies to models which have been pruned, quantized, distilled, and optimized to be ran in limited resource environments.

## 7  CONCLUSION

In this paper we explored different ways to customize a code generation model for a given codebase, with the goal of improving its performances on a target project. We described and analyzed four customization strategies and applied them on 20 different software projects for the task of generating unit test cases. Specifically, we considered the following strategies: (i) *custom* fine-tuning, which allows all the model parameters to be tuned on the target project; (ii) *L-EO* fine-tuning, a lightweight training which freezes most of the model's parameters, tuning only embedding and output layers; (iii) *L-LDB* fine-tuning, a lightweight training which only tunes the last decoder block; (iv) *prefix* tuning, which keeps language model parameters frozen, but optimizes a small project-specific vector (prefix).

In our extensive empirical evaluation we found that all the customization strategies lead to significant model's improvements on a target project, in terms of both intrinsic and task-specific metrics, with the custom models adapting to the coding style of the target project. While there is no clear winner among the customization strategies, each approach can provide specific benefits in particular deployment scenarios.

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
