# OpenReview forum: "Exploring and Evaluating Personalized Models for Code Generation"
_ICLR.cc/2022/Conference — ICLR 2022 Submitted_

### Official Review · Reviewer_v2C4 · 2021-10-24

**Correctness:** 2
**Technical Novelty And Significance:** 1
**Empirical Novelty And Significance:** 2
**Recommendation:** 1
**Confidence:** 5

**Main Review:**

Strengths:
+ Welcome empirical study of fine-tuning strategies specifically for source code applications.
+ Good use of statistical analysis helps correct for small dataset.

Weaknesses:
- Empirical analysis is too narrow, focusing on a single task and dataset, as well as very small models.
- Technical contributions, such as the fine-tuning of a single decoder layer, appear very ad-hoc.
- Overall results provide no actionable new insight into the use of fine-tuning for software engineering.

Fine-tuning and other domain adaptation methods are rightfully becoming important subjects in machine learning driven software engineering research, so an empirical study of this process is relevant. However, the approach in this work is too narrow to provide much in the way of actionable insights. The analysis is based on a single paper and dataset, on which it considers mostly long-established fine-tuning methods. The results are exactly as expected; complete-model fine-tuning is somewhat slower, but effective; fine-tuning just one or two layers is a bit faster and slightly less accurate (depending on compute), etc. Indeed, much of the motivation is very generic to the motivation for, and problems with, fine-tuning in general. The use of statistics to avoid drawing unwarranted conclusions is a welcome addition and further reinforces that no new results are found.

In terms of the original motivation, which identifies key factors such as privacy and memory/compute footprint, the results do not provide a particularly satisfying conclusion. The analysis entirely considers other open-source projects, so it shines little light on fine-tuning behavior within corporate code-bases, which may well have much more different characteristics to the training data than this test set. And the compute results in Figure 4 suggests that simply fine-tuning the full model is already the best solution at around 1 PF-second, which translates to much less than a minute on virtually every commonly used GPU. That seems like a _very_ small cost for adapting to a new project, which somewhat undercuts the need for any other analysis here. On the other hand, arguable more product-relevant issues such as working memory usage and inference time/cost are not addressed by this work.

The main technical novelty in this work stems from the addition of fine-tuning a single decoder layer. This is based on an analysis that shows that higher layers are updated (rather) slightly more in terms of parameter change when fine-tuning the full model. This feels very ad-hoc; it falls well short of more grounded analyses of fine-tuning different layers such as in Yosinski et al. (NeurIPS 2014), and its motivation suffers from the clear problem that parameter changes in full-model fine-tuning are not necessarily indicative of which layer "needs" to be fine-tuned the most. Updates naturally cascade across layers, so the behavior in Figure 2 seems natural when updating an entire model. The results indeed do not show a substantial difference between this and updating another layer (embedding/output), except perhaps at very specific compute values, for which no significance test was applied.

Detailed Comments:
- Introduction: please add references to substantiate the claims in the first two paragraphs
- P2, top paragraph: "performances" -> "performance"
- P2: the first two bullet points in the introduction paint a very similar picture, mostly because of the second half of the first, which also essentially talks about privacy.
- The use of the word "Custom" throughout the paper is rather odd. For one, it is often used interchangably with "customized", but those evoke rather different ideas -- "custom" suggests a specially designed model for a new task, which does not resonate the contributions here. "Customized" should really just read "fine-tuned"; I see no reason for another term. The variant called "Custom fine-tuning" (2.2) would be better called something like "Full model fine-tuning".
- P3, last paragraph: "finetuining" -> "finetuning" (twice)
- Table 2/3: why use such long IDs, instead of simply numbering the projects for this paper?
- What motivated fine-tuning both embeddings and the output layer, besides following Lu et al. -- are these weights tied in this model?
- Figure 2 is rather hard to read, primarily in terms of both axes' labels. Please also explain "(Other, 1)"
- Comparing matching rates on abstracted code mostly makes sense in terms of ignoring identifiers, but not distinguishing between different objects of the same type means this metric does not capturing semantic correctness. Please clarify to what extent this impacts the results; ideally ablate this latter decision to better isolate the effect of just naming conventions.

**Summary Of The Paper:**

Software projects tend to differ strongly from one another in terms of coding style, which significantly reduces the efficacy of even large pretrained models in new project contexts. This work explores the practical performance of several fine-tuning strategies of a model trained for a software engineering task (test-case generation) to new project contexts. The results suggest that fine-tuning can substantially improve model performance, but the specific choice of fine-tuning strategy makes very little difference.

**Summary Of The Review:**

The contribution is very narrow, targeting a single (far-from broadly used) task and dataset, and involving a relatively small set of projects that are not representative of the type of proprietary code the work is motivated as supporting. The results provide no new information beyond what is generally understood about fine-tuning, and lack the kind of systematic rigor seen in related work. Technical novelty is low, consisting of a single new fine-tuning approach that echoes traditional analyses, the motivation for which is also rather questionable.

---

### Official Review · Reviewer_bTU6 · 2021-10-30

**Correctness:** 3
**Technical Novelty And Significance:** 1
**Empirical Novelty And Significance:** 1
**Recommendation:** 3
**Confidence:** 4

**Main Review:**

*Strengths*

- The paper is easy to follow in most parts

*Weaknesses*

 - No novelty in the approach: The paper just analyzes already existing and well known approaches on a pre-existing model, dataset and task. The metrics used in the paper are pretty standard as well and there is no novelty at that end.

- No interesting insights: I find no interesting conclusions drawn from the analysis experiments. There was nothing discovered in the paper that is already not well known ( atleast for NLP). Just translating the same experimental settings and methods for one specific task and one model in code generation and not resulting in any interesting insights makes a very weak case for acceptance for me, keeping in view the ICLR standards.

- Lacking thorough evaluation: If empirical analysis is the only contribution, then I expect authors to be more thorough and extensive in terms of the following:
 i) considering more tasks like bug repair or code autocompletion
ii) considering more datasets and programming languages per task
iii) considering different models of source code especially the large language models like Codex

- Other questions/ concerns:
i) Is tf-idf similarity the best way to measure coding styles? I will suggest something that uses the parse trees and measure node-wise similarity. I am also not sure that measuring the tf-idf based on the identifiers from the unit tests previously written by the developers and the ones that are used in the test project say much about coding styles. The identifiers may be named purely based on the project for which the unit test case is written rather than the style of the developer.
ii) "On the bright side, this strategy allows to batch together requests from different users (with different prefixes), which can be processed by a single model, generating personalized outputs." I am not sure how exactly will this work.
iii) Even though it is stated in the abstract, introduction and conclusions that the authors specify guidelines as to when to use a particular strategy they don't discuss this explicitly anywhere in the paper. I was expecting a section dedicated to just discussing the specific use cases where a particular finetuning strategy can be used.
iv) I feel the details about the dataset like Table 2 can be moved as part of the supplementary material. I'd give more space to the analysis rather than specific implementation details.

**Summary Of The Paper:**

The paper analyses three types of fine-tuning approaches: (a) custom, (b) lightweight and (c) prefix on AthenaTest model for the task of unit test case generation in Java. The paper report results for 20 test projects in teens of BLUE, perplexity, task-specific performance as well as training cost computation.

**Summary Of The Review:**

No significant insight and not thorough experiments from an analysis paper point of view.

---

### Official Review · Reviewer_BJx7 · 2021-11-02

**Correctness:** 3
**Technical Novelty And Significance:** 1
**Empirical Novelty And Significance:** 1
**Recommendation:** 3
**Confidence:** 5

**Main Review:**

The paper compares the standard strategy of fine-tuning of all weights against two lightweight strategies and the prefix tuning strategy of (Li and Liang 2021). The lightweight strategies include fine-tuning only the embedding and output layers (Lu et al. 2021) and fine-tuning the last decoder layer.

These strategies are evaluated on the task of generating Java unit tests on 20 different projects, where the personalized models outperform the baseline model on BLUE4, perplexity and task-specific measures. The performance of various personalization strategies is mixed across the projects, with the standard strategy of fine-tuning all weights being the best overall. Thus, the paper does not offer any new dominant strategy within this space. It is unclear whether these observations hold in general for code generation as the evaluation is performed only on a single task.

The motivation of the paper is clear, but the proposed method of project-wise fine-tuning is expensive and difficult to scale in practice. Alternative methods such as few-shot learning have been shown to be successful for customizing large pre-trained models in the literature. These methods do not incur the cost of re-training. The paper does not evaluate the proposed strategies against them.

The assumption of having several hundred to thousand labeled examples for each project is also not practical. In contrast, few-shot learning usually requires only a handful of examples (something like 1-5).

**Summary Of The Paper:**

The paper proposes personalization of a baseline model to a project by fine-tuning it on the project-specific labeled examples. It demonstrates that personalization yields performance benefits over the baseline model. The baseline model (Tufano et al. 2021) is obtained from a pre-trained Java BART model after fine-tuning for the task of generating Java unit tests. The paper evaluates four fine-tuning strategies that select different subsets of weights to be fine-tuned.

**Summary Of The Review:**

The paper has limited technical novelty and empirical contributions. The project-wise fine-tuning strategy is not practical. There is no comparison against the few-shot learning techniques which can customize a model with a smaller number of examples and without re-training.

---

### Official Review · Reviewer_1jM4 · 2021-11-04

**Correctness:** 3
**Technical Novelty And Significance:** 1
**Empirical Novelty And Significance:** 2
**Recommendation:** 3
**Confidence:** 5

**Main Review:**

**Strengths:**
- The paper studies a relevant problem of customizing models for code towards individual/group/project preferences.
- The domain of personalized unit test case generation is novel, and could be useful for the broader code generation research community.
- Insights from the paper have, in my opinion, applications to settings beyond the specified server-side customization. Insights from RQ3, for example, can be useful for clients fine-tuning the model on their local devices to optimize for compute time and cost.

**Weaknesses:**
- This paper is mainly an empirical study of the effectiveness of various model fine-tuning strategies on the performance of the fine-tuned model. As an empirical study, I found the following things lacking:

    - **Evaluation:** This work utilizes BLEU score, Perplexity score, and Exact Match (EM) scores to evaluate the models. My concern here is that BLEU score is designed primarily for natural language evaluation and has been shown to not be a good measure for code evaluation [1]. Exact Match is not a good measure either as it simply checks for string equivalence while a piece of code can be written multiple ways and achieve the exact same goal. Why did the authors not choose to evaluate the generated test cases through execution, or the ability to compile as [2] does, or for their code coverage metrics? The paper does not touch upon these methods of evaluation, which I would argue are more important to judge the quality of generated test cases from an algorithmic and user point of view.

   - **Conclusion:** The conclusion of the study are:
       - All fine-tuning approaches improve the baseline model - this follows naturally from the definition of fine-tuning. Modifying the model weights over specific data distribution will improve the performance over a similar distribution during test time
      - Different customization strategies have different pros and cons: This is a trivial observation from how the customization strategies are defined. Modifying fewer weights will require fewer different sets of weights to be stored for customization.
     - All customization strategies achieve the same performance albeit with different levels of compute: This is the only insight that I found somewhat interesting, and something that could be useful to practitioners.

**Other clarifications:**

- Figure 2: There is no information on how this figure is created. It is mentioned as a motivation for L-LDB (Section 2.4), but does not describe what dataset/project that particular model is trained on, for how many iterations, and what the resulting performance is. Having experimental details of the procedure behind Figure 2 are important to understand the conclusions.

- Section 3.1 (Dataset): The authors state that they sample 20 project from the test set to fine-tune the model. They do not state what test set are these 20 projects subsampled from? Is it preexisting test set they utilize for the task, or they create a new dataset from GitHub? How are the parallel dataset of function <--> unit test created? Can the authors please provide more details on this.

- How is the *Abstracted Code Matches* metric computed? The paper simply mentions the conversion of code to an abstract form using *src2abs* tool, but does not discuss how the abstracted forms are matched thereafter.

- Section 3.4: The authors state that *"Backward pass takes roughly twice the amount of compute, but it is unnecessary for frozen layers"*. This is only true for L-LDB setting where once the final Decoder layer is fine-tuned, the backward propagation can be terminated. For all other cases, backward propagation needs to be done and the computation graph needs to be computed until the last trainable parameter, which is generally the entire network.

- The metric for compute (PF-seconds) unit is not defined or prior work utilizing/defining this metric cited anywhere.

[1] Hendrycks, Dan, et al. "Measuring Coding Challenge Competence With APPS." arXiv preprint arXiv:2105.09938 (2021).

[2] Tufano, Michele, et al. "Unit Test Case Generation with Transformers." arXiv preprint arXiv:2009.05617 (2020).

**Summary Of The Paper:**

- This work studies customizing models for code towards specific projects/coding standards/preferences for unit test case generation task. It specifically studies this in the context of server-side customization, where one entity would need to maintain multiple customized models, as opposed to client-side customization where each client can store its own customized model.
- It is an empirical study of how 4 customization strategies -- a) modify entire model weighs, b,c) modify a chosen subset of model weights, and d) prefix tuning -- affects the performance of the fine-tuned model on the test case generation task.
- This work finds no substantial difference in the final fine-tuned performance of any of the customization strategies, but argues that each strategy has its advantages and disadvantages, therefore allowing the user/implementor of this approach to make a more informed choice.

**Summary Of The Review:**

Overall, I found the problem well motivated but the empirical study and conclusions a little underwhelming. The paper could use A) better evaluation, B) More concrete conclusions and more discussion on when a particular technique might be useful over others (maybe experimental design imitating a particular situation to demonstrate the effectiveness of one approach over another could help), C) More qualitative and quantitative analysis into the test cases generated from various models, and D) Adding text to address concern I raised in the main review (dataset origins and collection procedure, compute metric description, etc.)

---

### Decision · Program_Chairs · 2022-01-20

**Decision:**

Reject

**Comment:**

The paper presents an empirical study of different strategies for fine tuning a large language model for the task of generating Java Unit tests *for a specific project*.

As several reviewers pointed out, the setup itself is fairly impractical, requiring fine-tuning on an individual project, thus making it applicable only to the very tail-end of very large projects where the investment of doing this would make sense and where one could reasonably collect sufficient data for that project.

On top of that, the paper contributes relatively little in terms of novel techniques. This in itself would be OK if the paper presented some extremely important empirical evidence. However, reviewers also raised some important concerns with the empirical evaluation itself. For example, as reviewer 1jM4 pointed out, there is prior research explicitly showing that the BLEU score is not a good measure for code evaluation.

Overall, the meta-reviewer agrees with the reviewers that this paper is below the bar for publication.